

# Effectiveness of specific stabilization exercise compared with traditional trunk exercise in women with non-specific low back pain: a pilot randomized controlled trial

Eduard Minobes-Molina[1], Maria Rosa Nogués[2], Montse Giralt[2], Carme Casajuana[2], Dyego Leandro Bezerra de Souza[3], Javier Jerez-Roig[1] and Marta Romeu[2]

[1] Research group on Methodology, Methods, Models and Outcomes of Health and Social Sciences- (M$_3$O), Faculty of Health Sciences and Welfare, Centre for Health and Social Care Research (CESS), University of Vic-Central University of Catalonia (UVIC-UCC), Vic, Barcelona, Spain
[2] Department of Basic Medical Sciences, Rovira i Virgili University, Reus, Tarragona, Spain
[3] Department of Collective Health, Federal University of Rio Grande do Norte, Natal, Rio Grande do Norte, Brazil

Corresponding authors
Eduard Minobes-Molina,
eduard.minobes@uvic.cat
Maria Rosa Nogués,
mariarosa.nogues@urv.cat

## ABSTRACT

**Background**. Non-specific low back pain (LBP) is the leading cause of disability worldwide. The primary physiotherapeutic treatment for LBP is physical exercise, but evidence suggesting a specific exercise as most appropriate for any given case is limited.
**Objective**. To determine if specific stabilization exercise (SSE) is more effective than traditional trunk exercise (TTE) in reducing levels of pain, disability and inflammation in women with non-specific low back pain (LBP).
**Design**. A pilot randomized controlled trial was conducted in Rovira i Virgili University, Catalonia.
**Methods**. Thirty-nine females experiencing non-specific LBP were included in two groups: the TTE program and SSE program, both were conducted by a physiotherapist during twenty sessions. The primary outcome was pain intensity (10-cm Visual Analogue Scale). Secondary outcomes were disability (Roland Morris Disability Questionnaire), and inflammation (IL-6 and TNF-$\alpha$ plasma levels). Measurements were taken at baseline, at half intervention, at post-intervention, and a month later.
**Results**. Mean group differences in change from baseline to post-intervention for TTE were: −4.5 points (CI 3.3 to 5.6) for pain, −5.1 points (CI 3.0 to 7.3) for disability, 0.19 pg/mL (95% CI [−1.6–1.2]) for IL-6 levels, and 46.2 pg/mL (CI 13.0 to 85.3) for TNF-$\alpha$ levels. For SSE, differences were: −4.3 points (CI 3.1 to 5.6) for pain, −6.1 points (CI 3.7 to 8.6) for disability, 1.1 pg/mL (CI 0.0 to 2.1) for IL-6 levels , and 12.8 pg/mL (95% CI [−42.3–16.7]) for TNF-$\alpha$ levels. There were an insignificant effect size and no statistically significant overall mean differences between both groups.
**Conclusion**. This study suggests that both interventions (traditional trunk and specific stabilization exercises) are effective in reducing pain and disability in non-specific LBP patients, but the two programs produce different degrees of inflammation change.
**Clinical trial registration number**. NCT02103036.

## INTRODUCTION

Common low back pain (LBP) is defined as pain between the costal margins and the inferior gluteal folds, usually accompanied by painful limitation of movement, often influenced by physical activities and posture, and which may be associated with referred pain in the leg (*Kovacs et al., 2006*). LBP represents an important public health problem, because worldwide prevalence of the condition ranges from 12% to 33% (*Walker, 2000*). It is also known that LBP is more prevalent in females than males; for example, in 2015 in Catalonia, 30.1% of females suffered LBP, as compared to 18.7% of males (*Garcia, Medina & Schiafinno, 2016*). LBP remains a common disabling condition (*Walker, Muller & Grant, 2004*) and is associated with high costs for medical health and social care (*Haldeman et al., 2012*; *Wieser et al., 2011*).

One of the difficulties in reducing the burden of spinal disorders is the wide and heterogeneous range of specific diseases and non-specific musculoskeletal disorders that can involve the spinal column, most of which manifest pain (*Haldeman et al., 2012*). Despite this factor, or perhaps because of its impact on individuals, their families, and the healthcare systems, spinal disorders remain one of the most controversial and challenging conditions (*Haldeman et al., 2012*; *Hoy et al., 2010*).

Back pain is sometimes associated with a likely aetiology (e.g. radiculopathy or spinal stenosis), but most LBP cases are of unknown origin and are classified as non-specific, which has also been described as mechanical pain or strain, account for 90% or more of all people experiencing spinal pain (*Haldeman et al., 2012*; *Van Tulder et al., 1997*). For all these reasons, it is necessary to find an effective treatment for LBP. Unfortunately, the scientific literature does not offer relevant conclusions, due in part to the poor methodology employed in many published studies—e.g., short follow-up periods, population heterogeneity and non-validated measurements (*Atlas & Nardin, 2003*).

According to current clinical reviews and guides, first-line treatments for LBP pathology focus on analgesic measures (*Koes et al., 2001*; *Van Tulder et al., 2006*). Conservative physiotherapeutic treatments for LBP also exist (*Bordas et al., 2004*), including advice and postural education, electrotherapy, manual therapy, and physical exercises (*Williams et al., 2014*). Exercise is one of the chief recommendations for pain reduction, mobility increase, improvement of physical and psychological abilities and anxiety reduction (*Waddell & Burton, 2005*). The problem with exercise programs lies in the fact that rationales for choosing the appropriate exercise for an individual case are very weak. Controversy arises because the types of exercise programs for LBP vary considerably, as do the types of patients. This makes it very unlikely that a particular program will be equally effective in all cases (*Macedo et al., 2008*).

One exercise option is the back school program, a therapeutic program including information on the anatomy of the back, biomechanics, optimal posture, ergonomics, and back exercises (*Parreira et al., 2017*), which has proven effective in reducing pain and

disability (*Sahin et al., 2011*). There are many types of back school exercise programs, but considerable discussion has centered on the question of whether the specific stabilization exercise program (SSEP)—in which the deep muscles are the protagonists—is preferable to the traditional trunk exercise program (TTEP), which includes more exercises for strengthening abdominal and back muscles. There are systematic reviews that support the idea that SSEP is superior to TTEP (*Lederman, 2010*), but there are also studies that have found both approaches equally effective for improvement of LBP in terms of pain and disability (*Shamsi, Sarrafzadeh & Jamshidi, 2015*).

On the other hand, pro-inflammatory cytokines levels were detected on assessing local tissue in adults with LBP (*Queiroz et al., 2015*) and evidence shows that physical exercise therapy decreases systemic inflammatory mediators production, this demonstrates its clinical relevance (*Pereira et al., 2013*).

After years of research into LBP treatment, and taking into account the variety of treatment options, exercise continues to be accepted as an effective approach. The question remains, however, which type of exercise is most effective in treating various patient subgroups (*Atlas & Nardin, 2003*; *Saner et al., 2011*). There is limited evidence that the specific stabilization exercise program is more effective than the traditional trunk exercise for patients with non-specific LBP. Therefore, the following were the research questions this study sought to answer:

1.  Is the specific stabilization exercise program more effective than the traditional trunk exercise program in reducing levels of pain and disability in women with non-specific LBP?
2.  Which type of back school exercise produces different degrees of inflammation change in women with non-specific LBP?

## MATERIALS & METHODS

### Ethics

The Clinical Ethics Committee of University Hospital Sant Joan of Reus approved this study (12-06-28/6proj4). All participants gave written informed consent before data collection began.

### Study design

A pilot randomized trial was conducted in Catalonia from February 2013 to February 2015 (NCT02103036). Participants, diagnosed by medical practitioners and referred for treatment of non-specific LBP, were randomized using computer-generated random number tables into two treatment groups: a TTEP group and a SSEP group (Fig. 1). Afterwards, measurements were taken at baseline (session 0), at half intervention (session 10), post-intervention (session 20) and one month later. A single-blind study was conducted, due to the impossibility of achieving double blindness—the physiotherapist performing the intervention had to know which treatment each participant was to receive. The therapist who performed the intervention was a qualified health professional with 5 years' experience in the field; a different professional took all the study measurements,

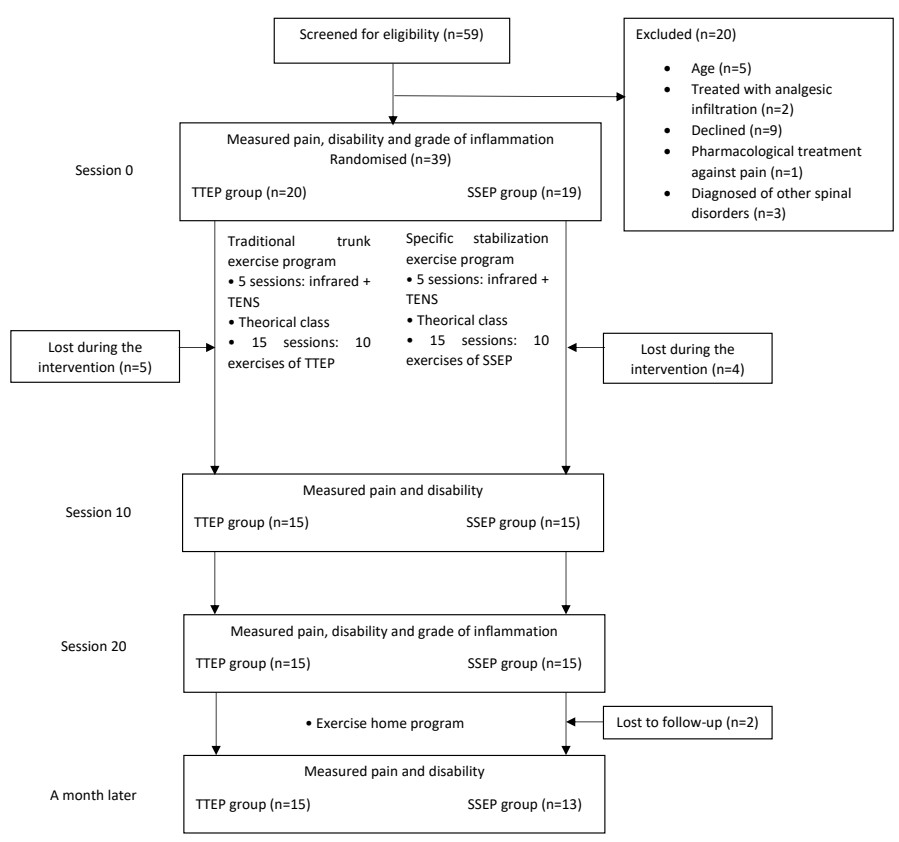

**Figure 1** Design and flow of participants through the trial.

he was blinded to the participant's assignment. Participants were blinded to their group allocation, design and hypotheses (*Page & Persch, 2013*).

## Participants

Participants entering the trial were required to meet the following inclusion criteria: females aged between 18 and 70 years; diagnosed with non-specific LBP (fewer than 6 weeks of pain duration) by a specialist doctor who used imaging such as magnetic resonance, radiographic or computed axial tomography to rule out other spinal disorders, and under no pharmacological treatment for pain. Exclusion criteria were: diagnosed with other spinal disorders and/or any other serious co-morbidities (e.g., cancer, severe lung pathology); presence of cognitive impairment; inability to perform exercises; having followed a specific training program with a physiotherapist in the previous three months; having been treated with analgesic infiltration in the previous 6 weeks, or failure to follow their 20-treatment schedule exactly. All study participants were volunteers, and all underwent intervention under the Faculty of Medicine and Health Sciences of University Rovira i Virgili.

## Intervention

Both treatment groups (TTEP and SSEP) underwent 20 sessions of treatment at a frequency of three to five sessions per week (*Saner et al., 2011*), as follows:

In the first five sessions, the only treatments were application of an infrared lamp and transcutaneous electrical nerve stimulation (TENS), since these have been proven to reduce pain in both acute and chronic LBP (*Bertalanffy et al., 2005*; *Jauregui et al., 2016*; *Van Tulder et al., 2000*). Patients did not perform any exercise in these first sessions, since there is strong evidence that exercise is not effective in relieving acute pain, and can even worsen symptoms (*Van Tulder et al., 2000*). TENS therapy was applied with a multichannel portable TENS unit (Megasonic 313 P4, Carin) on the lumbar spine. Biphasic square wave impulses at a frequency of 100 Hz and pulse duration of 70 μs were used for a total duration of 20 min. Four rectangular $90 \times 45$-mm electrodes were applied on the fascia thoracolumbaliis and approximately 10 cm proximal to this, along the midline of the muscle (i.e., directly over the site of pain) (*Kofotolis, Vlachopoulos & Kellis, 2008*).

In sessions six through 20, each group engaged in its respective back school exercise program in regular sessions (Fig. 2); they followed 30-minute protocols of 10 exercises, with 10 repetitions of each. The TTEP group performed exercises from the LBP protocol developed by the Physiotherapy and Rehabilitation Service at Sant Joan University Hospital (Reus, Spain). The SSEP group performed exercises gathered in a search of literature on core stability exercises (*Koumantakis, Watson & Jacqueline, 2005*; *Shamsi, Sarrafzadeh & Jamshidi, 2015*). Before beginning exercises, participants received education on the anatomy of the back (members of the SSEP received a simplified explanation of core musculature), correct posture, and spinal alignment, as part of a back school.

The same physiotherapist supervised all classes. Upon treatment completion, he gave each participant a home exercise program which included the exercises from their treatment sessions.

## Outcome measurements

The primary outcome was pain, measured with a 10-cm Visual Analogue Scale (VAS), which has been shown valid and reliable. VAS is a numerical rating scale (0 = no pain to 10 = worst imaginable pain) which represents the intensity of the current pain and allows the evaluator to compare it with previous or later evaluations (*Hawker et al., 2011*). VAS was used to measure pain at baseline session 0 and at sessions 10 and 20. Pain measurement one month after the final treatment session was done by telephone, using the 11-point Numerical Rating Scale (NRS). During this phone conversation, each patient also performed a verbal, subjective assessment. Both these final assessments are considered sufficiently sensitive to detect clinically relevant pain changes (*Hawker et al., 2011*); they are even considered interchangeable for calculating pain in lumbar pathologies (*Hawker et al., 2011*; *Thong et al., 2018*; *van Tubergen et al., 2002*).

One secondary outcome was disability, measured using the Roland Morris Disability Questionnaire (RMDQ). The questionnaire asks 24 questions related to the participant's current functional status. Different studies have shown RMDQ to be a useful and reliable instrument for evaluating participants with LBP (*Payares, Lugo & Restrepo, 2015*). RMDQ

| | Traditional trunk exercise<br>Muscles involved: global abdominal<br>and back muscles | Specific stabilization exercise<br>Muscles involved: deep muscles<br>(transversus abdominis, multifidus and<br>internal oblique muscles) |
|---|---|---|
| 1 | | |
| 2 | | |
| 3 | | |
| 4 | | |
| 5 | | |
| 6 | | |
| 7 | | |
| 8 | | |
| 9 | | |
| 10 | | |

**Figure 2    Progress of the exercise programs.**

was used to measure pain at baseline session 0, at sessions 10 and 20, and one month after the final treatment session.

Another secondary outcome was degree of inflammation, as measured by blood-sample levels of the cytokines interleukin 6 (IL-6) and tumor necrosis factor alpha (TNF-$\alpha$). These markers were used because they have been found to play significant roles in relation to back pain (*De Queiroz et al., 2016*; *Kraychete et al., 2010*). The presence of some inflammatory
mediators might be associated with pain and disability in patients with LBP, since pro-inflammatory cytokines such as IL-6 or TNF-$\alpha$ contribute to the activation of nociceptors that generate potential of action and pain hyper sensibility (*Cui et al., 2000*; *Queiroz et al., 2015*). Degree of inflammation was measured at the beginning and end of the treatment program by enzyme-linked immunosorbent assay (ELISA) method. Blood samples were collected by a qualified doctor (blinded to group allocation) and will be obtained from the antecubital vein (*Tomazoni et al., 2019*).

The study recorded additional factors, including anthropometric characteristics (age, height, weight, body mass index [BMI]), and degree of physical activity using Quick Classifier of Physical Activity (ClassAF) in Metabolic Equivalent of Tasks (METs). ClassAF is a global questionnaire which classifies people as physically active or inactive using a corresponding qualitative formula (*Vallbona et al., 2007*). All these data were collected before the intervention began by the trained physiotherapist.

## Data analysis

Groups were compared with respect to change, from baseline (session 0) to half-intervention (session 10), baseline to post-intervention (session 20), and baseline to 1 month after the intervention concluded; from session 10 to session 20 and session 10 to one month post intervention; and finally from session 20 to one month post intervention.

SPSS program version 23 Windows was used to analyze the data. A descriptive analysis was made of the study sample, with standard averages, deviations and percentages of the different variables collected. The Kolmogórov-Smirnov test was applied to assess data distribution in each group. A Student-t test was done to assess differences between the two treatments, and effect size was calculated to measure the magnitude of the experimenter effect, using the standardized mean difference (SMD) for variables normally distributed and the effect size of Mann–Whitney's U test for variables not normally distributed (*Field, 2005*). Two-way repeated ANOVA analyses were used to examine differences over time. Assessments were carried out using non-parametric tests for variables that did not present normal distributions. The level of statistical significance for the study was established at $p < 0.05$.

## RESULTS

### Flow of participants and therapists through the trial

59 potential participants were referred to the research team. Of those referred, 20 were not included, for various reasons (Fig. 1). 20 participants were placed in the traditional trunk exercise group (TTEP); the remaining 19 were placed the specific stabilization exercise group (SSEP). Of these 39 participants, 30 completed their course of treatment (15 from each group). Table 1 shows participants' baseline characteristics: age, height, weight, BMI, physical activity level, pain or disability. According to the CONSORT statement, significance testing of baseline differences in randomized controlled trials were not performed (*Moher et al., 2010*).Two members of the SSEP group could not be reached for the one-month follow-up telephone call.

**Table 1 Baseline anthropometric and clinical characteristics of the participants.**

| Characteristic | Total ($n = 30$) | TTEP Group ($n = 15$) | SSEP Group ($n = 15$) |
|---|---|---|---|
| Age (yr), mean (SD) | 50.5 (10.2) | 50.9 (11.0) | 50.1 (9.8) |
| Height (m), mean (SD) | 1.6 (0.1) | 1.6 (0.1) | 1.6 (0.1) |
| Weight (kg), mean (SD) | 68.8 (9.8) | 66.8 (9.4) | 70.9 (10.0) |
| BMI (kg/m$^2$), mean (SD) | 26.9 (3.6) | 26.3 (3.5) | 27.5 (3.7) |
| Grade of physical activity (METS), mean (SD) | 13.5 (19.0) | 6.8 (6.3) | 19.7 (24.5) |
| Pain (cm), mean (SD) | 6.5 (1.4) | 6.4 (1.2) | 6.5 (1.6) |
| Disability (points), mean (SD) | 9.2 (3.9) | 8.9 (4.1) | 9.5 (3.9) |

**Notes.**
TTEP, traditional trunk exercise program; SSEP, specific stabilization exercise program; BMI, body mass index; MET, metabolic equivalent of task.

**Table 2 Mean (SD) for outcomes reported at all study visits for total and each group, significant differences between visits within groups, $p$ values, mean difference (95% CI) and effect size (95% CI) between groups for pain intensity and disability.**

| Clinical outcome | | Total ($n = 30$) | TTEP Group ($n = 15$) | SSEP Group ($n = 15$) | $p$ value TTEP- SSEP | Mean difference (95% CI) | Effect size (95% CI) |
|---|---|---|---|---|---|---|---|
| Pain (0–10 cm) | VAS[1] | 6.5 (1.4) | 6.4 (1.2) | 6.5 (1.6) | 0.80 | −0.13 (−1.20–0.93) | −0.09[NND] (−0.45–0.27) |
| | VAS[2] | 3.2[a](2.0) | 3.3[a](1.7) | 3.1[a](2.3) | 0.79 | 0.20 (−1.32–1.72) | 0.10[ND] (−0.26–0.46) |
| | VAS[3] | 2.0[a,b] (1.8) | 1.9[a,b] (1.7) | 2.2[a,b] (1.9) | 0.62 | −0.33 (−1.67–1.01) | −0.19[NND] (−0.55–0.18) |
| | NRS[4] | 2.8[a](3.0) | 2.8[a](3.1) | 2.9[a](3.0) | 0.97 | −0.05 (−2.40–2.31) | −0.02[NND] (−0.37−−0.34) |
| Disability (0-24 points) | RMDQ[1] | 9.2 (3.9) | 8.9 (4.1) | 9.5 (3.9) | 0.68 | −0.6 (−3.57–2.37) | −0.15[ND] (−5.12–0.21) |
| | RMDQ[2] | 5.0[a](3.2) | 5.3[a](3.8) | 4.8[a](2.7) | 0.70 | 0.47 (−1.99–2.93) | 0,15[ND] (−0.22–0.50) |
| | RMDQ[3] | 3.6[a,b](2.8) | 3.8[a,b] (3.3) | 3.4[a,b] (2.2) | 0.70 | 0.40 (−1.72–2.52) | 0.14[NND] (−0.28–0.50) |
| | RMDQ[4] | 4.3[a](4.3) | 3.9[a](3.8) | 4.8[a](4.9) | 0.62 | −0.84 (−4.22–2.55) | −0.20[NND] (−0.56–0.17) |

**Notes.**
TTEP, traditional trunk exercise program; SSEP, specific stabilization exercise program; VAS, visual analogue scale; NRS, numerical rating scale; RMDQ, Roland Morris disability questionnaire; CI, Confidence Interval; ND, Normal distribution; NND, Not normal distribution.
$p < 0.05$= significant difference between groups.
[1]Baseline (session 0).
[2]Half intervention (session 10).
[3]Post-intervention (session 20).
[4]A month post-intervention.
[a]Significant difference from session 0.
[b]Significant difference from session 10.

## Compliance with the trial method

30 (76,9%) participants attended all 20 intervention sessions. Once the intervention was completed, the physiotherapist advised participants to repeat their exercises at home, three times a week for one month. 15 (50%) participants reported performing the exercises as advised; 9 (30%) reported doing their exercises occasionally; 4 (13.3%) did not perform exercises at home; the remaining 2 (6,67%) were unreachable.

## Effect of intervention

Data on pain and disability are shown in Table 2; data on degree of inflammation are in Figs. 3 and 4.

Results show an insignificant effect size and no significant differences between groups in terms of current pain intensity, or for any outcome measure. At the end of intervention (session 20), pain intensity for the TTEP group had decreased by 0.33 cm (95% CI

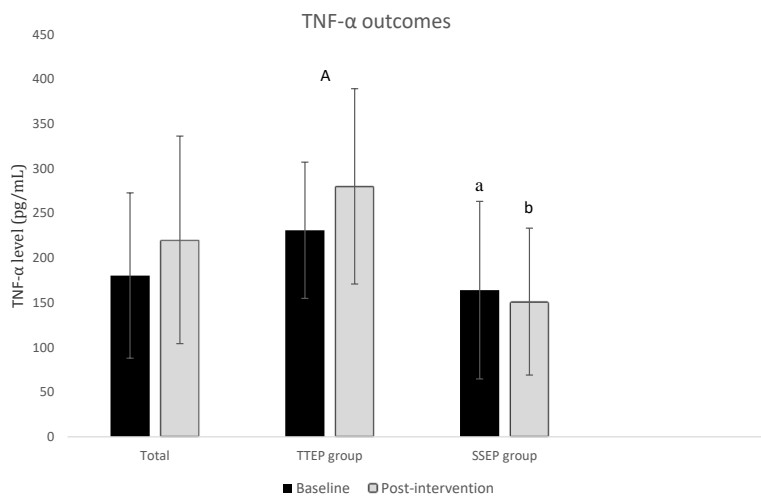

**Figure 3  Mean (SD) TNF-α biomarker levels in TTEP group and SSEP group at baseline (session 0) and post-intervention (session 20).** TNF-α, tumour necrosis factor alpha; TTEP, traditional trunk exercise program; SSEP, specific stabilization exercise program; A, significant difference between baseline and post-intervention in TTEP group; a, significant difference between baseline groups; b, significant difference between post-intervention groups.

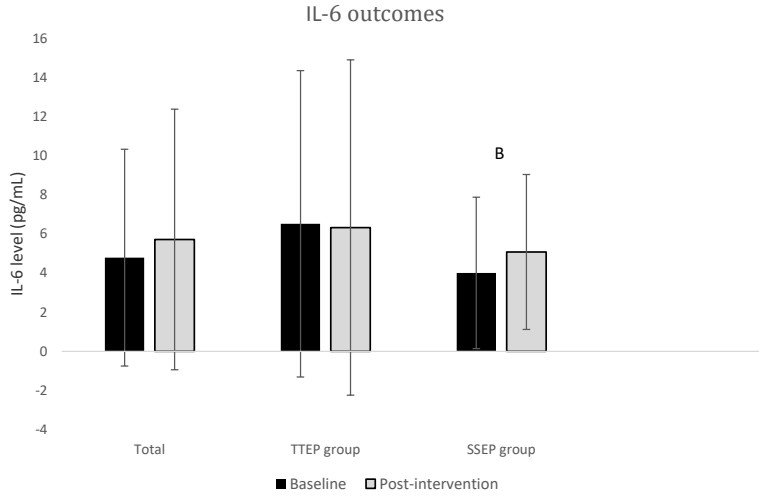

**Figure 4  Mean (SD) IL-6 biomarker levels in TTEP group and SSEP group at baseline (session 0) and post-intervention (session 20).** IL-6, interleukin 6; TTEP, traditional trunk exercise program; SSEP, specific stabilization exercise program; B, significant difference between baseline and post-intervention in SSEP group.

$[-1.7–1.0]$, $p = 0.615$) more than in the SSEP group. Both back school treatments showed positive results for pain reduction from baseline to end of treatment, and baseline to one month post-intervention. In the TTEP group, pain (baseline to final session) reduced by 4.6 cm (95% CI [3.3–5.8]); the SSEP group's reduction was 4.3 cm (95% CI [3.0–5.6]).

Similarly, there were an insignificant effect size and no significant differences between groups in terms of change in disability. At post-intervention (session 20), disability levels in the SSEP group had decreased by 0.40 points (95% CI [−1.7–2.5], $p = 0.701$) more than the TTEP group. Both back school treatments yielded positive results in disability reduction, baseline to end of treatment, and baseline to one month post-treatment. In the TTEP group, RMDQ scores reduced by 5.1 points (95% CI [3.0–7.3]) from baseline to post-intervention. In the SSEP group, RMDQ reduction for the same interval was 6.1 points (95% CI [3.7–8.6]).

Figures 3 and 4 show outcomes for inflammation. TNF-$\alpha$ showed higher values for the TTEP group than the SSEP group at the two visits where TNF-$\alpha$ was measured. Significant differences were observed between the groups, baseline and post-treatment. In the first case, a difference of 66.97 pg/mL (95% CI [6.3–139.5]) was recorded; in the second case the difference was 128.94 pg/mL (95% CI [52.8–205.0]).

In contrast, IL-6 levels were found to be similar between the two treatment groups, with no significant differences observed. At baseline the difference was 2.51 pg/mL (95% CI [−2.3–7.3]); at post-intervention the difference was 1.25 pg/mL (95% CI [−3.9–6.4]).

In reference to the evolution of inflammatory biomarkers between baseline and post-treatment, the results for participants who practiced traditional TTEP indicate an increase in TNF-$\alpha$ levels of 46.16 pg/mL (95% CI [13.0–85.3]) and a tendency toward decreased levels of IL-6, 0.19 pg/mL (95% CI [−1.6–1.2]).

In contrast, the results in the group that practiced SSEP are the other way around: there was an increase in IL-6 levels of 1.06 pg/mL (95% CI [0.03–2.1]) and a tendency toward decrease in TNF-$\alpha$ levels of 12.81 pg/mL (95% CI [−42.3–16.7]).

## DISCUSSION

Our study hypothesis suggested that treatment with SSEP would be found to decrease pain and disability more effectively than TTEP, in women with non-specific LBP. We found this hypothesis not entirely true—although the effectiveness of SSEP was apparently demonstrated, the effectiveness of TTEP was found to be quite similar in our study group.

The literature documents study results confirming those of our own study: *Shamsi, Sarrafzadeh & Jamshidi (2015)* also concluded that the two types of exercise provide improvement in LBP, but found no evidence as to which type might be more effective.

The literature also includes meta-analyses comparing back schools for chronic LBP. These found deep-muscle exercises more effective in reducing short-term pain and disability (*Chang, Lin & Lai, 2015*; *Niederer & Mueller, 2020*; *Wang et al., 2012*), though they found no significant differences in long-term improvement. In our study, however, after the half-way point in treatment (session 10) we observed pain reduction by both modalities. We believe this was most likely due to the fact that participants in the meta-analyzed clinical trials suffered from chronic lower back pain, which has a worse prognosis than non-specific LBP in an early phase (*Van Den Hoogen et al., 1998*).

Contrary to our results, in a recent systematic review, a meta-analysis of 8 studies indicated that stabilization exercises were more effective than general exercises in reducing

pain. Five studies demonstrated a significant improvement in disability between patients treated with stabilization exercises compared with those treated with general exercises (*Gomes-Neto et al., 2017*). In our case, the SSEP and TTEP seem to be effective in reducing pain and improving disability. The mean of pain in the analyzed studies was 6.01 at baseline, being 2.1 at the end of the stabilization exercises on a 0-10 pain scale (*Gomes-Neto et al., 2017*). The SSEP results of our trial are consistent with these findings: 6.53 at baseline and 2.2 at the end of the intervention.

There are authors who have found that stabilizing treatment shows no significant advantage over traditional treatment (*Koumantakis, Watson & Jacqueline, 2005*); some of these authors believe that where there appears to be such an advantage, it is due to certain characteristics of the LBP patients involved, such as segmental instability of the column, or the size of the multifidus muscles. One of the exclusion criteria in our study was diagnosis of other spinal disorders, so our sample was more homogeneous.

Our results with regard to inflammation indicate that, following TTEP, TNF-$\alpha$ levels had increased; when SSEP was used, IL-6 levels had increased by the end of our 20-session course of treatment.

*Al-Obaidi & Mahmoud (2014)* recently reported on a study with characteristics similar to ours, which found increased production of pro-inflammatory cytokine TNF-$\alpha$ after treatment, but no change in IL-6 production. The author justified this result by stating that overexpression of TNF-$\alpha$ and other pro-inflammatory cytokines occurs in many studies of low-back pathologies (*Takahashi et al., 1996*). In addition, he explained, IL-6 cytokine levels are not altered because IL-6 has both pro-inflammatory and anti-inflammatory properties (*Opal & Depalo, 2000*).

Various studies claim that IL-6 acts predominantly as an anti-inflammatory cytokine, regulating the synthesis of pro-inflammatory cytokines IL-1 and TNF-$\alpha$ and stimulating the appearance, in circulation, of anti-inflammatory cytokines such as IL-10 (*Opal & Depalo, 2000*; *Saavedra Ramírez, Vásquez Duque & González Naranjo, 2011*). One study goes further (*Petersen & Pedersen, 2005*), claiming that IL-6 stimulates lipolysis and oxidation of fats, as well as producing anti-inflammatory effects during exercise—and therefore may offer protection against TNF-$\alpha$. Relating this information to our own findings, we could say that treatment with SSEP aims to be more effective because, in our case, TNF-$\alpha$ levels were maintained while IL-6 increased. On the other hand, with TTEP the reverse was true: the cytokine found to have increased in the plasma was TNF-$\alpha$.

We believe this is due to the nature of the exercises. In SSEP, deep muscle exercise is the basis of lumbar and segmental control stabilization. TTEP, on the other hand, focuses on building overall muscle resistance, strength and flexibility, being a more dynamic and intense activity. The literature includes findings that lower-intensity exercises are more effective than those of greater intensity, when it comes to reducing inflammation (*Ghafourian et al., 2016*).

One of our study's limitations is its sample size, but we also prioritized for this pilot trial the homogeneity of our patients through strict inclusion and exclusion criteria, for example we only studied women due their physiological characteristics such as less muscle and bone mass as well as psychological factors (*Hoy et al., 2012*). A study design with larger samples

would allow a greater effect size between groups and the creation of subgroups according to age, degree of physical activity, or BMI—facilitating more definitive conclusions regarding these factors. Further, we believe it would be interesting to add another follow-up, beyond this study's one-month-post-intervention evaluation. Further follow-up (at six months, for example) would reveal any difference between the treatments in terms of long-term clinical improvement, although the results of the current literature suggest that SSEP improves pain and functional status at 3 months but not at 6 or 12 months (*Coulombe et al., 2017*).

In summary, this study suggests that any type of back school exercise is highly effective in reducing pain and reducing disability in women with non-specific LBP. Further, it showed that SSEP seems to have an anti-inflammatory effect in such patients, potentially offering protection against chronic diseases associated with low-grade inflammation (*Petersen & Pedersen, 2005*).

## CONCLUSIONS

This study adds to the literature the finding that both back school exercise program are apparently effective and equivalent in reducing pain and improving disability in women with non-specific LBP, from the tenth treatment session to one month after intervention. Moreover, it demonstrates the influence of each back school in the degree of inflammation, concluding that SSEP seems to increase production of anti-inflammatory biomarkers, while TTEP increases pro-inflammatory biomarker production. A large, adequately powered study is recommended to determine if the results from this pilot study can be duplicated.

## ACKNOWLEDGEMENTS

We thank all patients for their participation in this study. Furthermore, we thank the physiotherapists Iris Alonso, Idoia Garitano and Cristina Pellicer for the collaboration.

### Funding
The authors received no funding for this work.

### Competing Interests
The authors declare there are no competing interests.

### Author Contributions

- Eduard Minobes-Molina conceived and designed the experiments, performed the experiments, analyzed the data, prepared figures and/or tables, authored or reviewed drafts of the paper, and approved the final draft.
- Maria Rosa Nogués and Marta Romeu conceived and designed the experiments, performed the experiments, analyzed the data, authored or reviewed drafts of the paper, and approved the final draft.
- Montse Giralt, Carme Casajuana and Javier Jerez-Roig conceived and designed the experiments, authored or reviewed drafts of the paper, and approved the final draft.

- Dyego Leandro Bezerra de Souza conceived and designed the experiments, analyzed the data, authored or reviewed drafts of the paper, and approved the final draft.

## Human Ethics

The following information was supplied relating to ethical approvals (i.e., approving body and any reference numbers):

The Clinical Ethics Committee of University Hospital Sant Joan of Reus approved this study (12-06-28/6proj4). This organization is associated with the Rovira i Virgili University, organization with 4 affiliated authors, where the intervention took place.

## Clinical Trial Ethics

The following information was supplied relating to ethical approvals (i.e., approving body and any reference numbers):

The Clinical Ethics Committee of University Hospital Sant Joan of Reus approved this study (12-06-28/6proj4). All participants gave written informed consent before data collection began.

## Data Availability

The raw measurements are available in the Supplemental Files.

## Clinical Trial Registration

The following information was supplied regarding Clinical Trial registration:

NCT02103036

## Supplemental Information

Supplemental information for this article can be found online at http://dx.doi.org/10.7717/peerj.10304#supplemental-information.

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
