# Peer review of "Effectiveness of specific stabilization exercise compared with traditional trunk exercise in women with non-specific low back pain: a pilot randomized controlled trial"

_PeerJ, doi:10.7717/peerj.10304_

## Round 0.1 · original submission · Major Revisions

The three reviewers and I have identified a number of important issues that the authors need to address before this manuscript can be more strongly considered for publication in PeerJ. Therefore, please make sure you look to address all of the comments from the three reviewers in your revised manuscript. In particular, due to the small sample size, it is suggested that the authors rephrase this as a pilot/feasibility study.

Reviewer 1 ·

Basic reporting

Please see "General Comments".

Experimental design

Please see "General Comments".

Validity of the findings

Please see "General Comments".

Additional comments

Thank you for the opportunity you gave me to review the manuscript entitled “Effectiveness of specific stabilization exercise compared with traditional trunk exercise in women with non-specific low back pain: a randomized controlled trial”. However, in my opinion, the manuscript needs to be improved to be published in PeerJ.

Specific comments
1- Materials & Methods: The authors may need explain the DETAILS of the randomization method (e.g., allocation concealment mechanism, implementation, and blinding).
2- The authors used p-values to assess balance in baseline characteristics between two groups. There are many published references that avoid researchers to use of statistical testing for baseline comparison (de Boer et al., ‎2015). P-values are affected by sample size. If the sample size is large enough, even very small differences may be statistically significant. On the other hand, even large differences may lead to non-significant results if the sample is too small (less than 20) (du Prel et al., 2009). For quantitative variables, other approaches such as difference of <0.25 standard deviation (SD) in baseline characteristics can be used.
3- The authors may need to calculate effect size measures (e.g., the standardized mean difference [SMD]) with associate confidence intervals in order to compare between two groups. The major benefit of determining effect size is that unlike test statistics, the effect size is not greatly influenced by sample size, thus reducing problems of power associated with large and small samples.
4- For variables that were not normally distributed, the authors can use transformation to calculate SMD.
5- The authors may need to present the significance level sign (*) in figures 2 and 3 for within-group comparison.
6- Is was not clear whether confounding variables (e.g., Smoke and depression) were controlled in patients with NLBP.
7- It is highly recommended that the authors compare the effect of intervention on the outcome measures with their MCID values to assess whether the effect is substantially beneficial.
8- In addition to statistical tests, the authors may need to assess the normal distribution with histograms.
9- The authors mentioned that “One of our study’s limitations is its sample size…” Why did the authors state this sentence whereas they calculated the sample size?
10- The authors may need to report of the mode of TENS (conventional, low rate, etc.) with its details (i.e., frequency, electrode placement, pulse width, time, etc).
11- How did the physiotherapist progress the exercise program?
12- The figures of exercises should be presented.
13- Did the participants use analgesics during the treatment period? If yes, how did the authors manage this issue?

Reviewer 2 ·

Basic reporting

The english writing is acceptable The literature review content is very cursory around the issue of comparing different exercise types, as the evidence is overwhelming that no one type of exercise is superior to another (incl stabilization exercise). There is next to no content to justify the inflammatory measures used and they should be removed moving forward. This article should be scoped as a feasibility study and described as such.

Experimental design

The risk of bias is unacceptable for this study with the single physiotherapist conducting both interventions, thus we have no faith either intervention is valid.

The sample size estimation to justify such a small sample is extremely flawed, and scoping the study as a feasibility study is best. However the single physiotherapist as the treating professional is unacceptable to have any confidence in the validity of the intervention.

Validity of the findings

This section is a fail as per the above critical error in study design.

Additional comments

This could be reframed as a clinical feasibility study, but the issue of a single clinician providing the two fundamentally different types of exercise is a major bias that cannot be corrected or avoided here. There can be no confidence in the validity of the interventions or the data.

Reviewer 3 ·

Basic reporting

The article in general uses clear, concise, unambiguous and technically correct text.
The introduction and background sufficiently describes the impact of LBP. Relevant prior literature is appropriately referenced.
The article structure, figures and tables are all of professional level, however some specific comments for the Results, Discussion and figures sections are provided below. Raw data were not shared.
The structure of the article conforms to an acceptable format of ‘standard sections’.
The article is generally self-contained with relevant results to hypotheses. However, some issues require a clearer explanation and should be added.

Experimental design

Original primary research with research question well defined, relevant & meaningful, stating how research fills an identified knowledge gap. It is interesting that the authors wished to explore inflammatory markers in LBP.

A well-conducted investigation performed to a sufficient technical & ethical standard.

Methods not described with sufficient detail & information to replicate in some respect, and this is pointed out to the authors.

Validity of the findings

The validity of the findings is not sound, as results are representative of a small group of LBP participants and only female. Therefore this study can be characterized as a pilot study.

Conclusions are adequately stated, linked to original research question & limited to supporting results.

Additional comments

1. In my opinion, TNF-a and IL-6 values in the Abstract should be made for both groups. Please amend.

2. Follow up of 1 month post-exercise is a rather limited timeframe. However, within that limited time, pain and disability levels in both groups have increased. Possible implications of a non-long-lasting effect of both exercise programs on pain & disability levels should be commented on by the authors.

3. The authors seem to over-discuss the significance of their secondary measures (not the Disability one) over their primary one.However, for such a small study, and probably underpowered for those particular measures, such discussion is somewhat redundant. For instance, TNF-a unequal baselines, is probably due to small sample size. On the other hand, IL-6 baseline levels were equal between-groups. Which renders this latter measure the one that confers an equal starting point. For TNF-a levels, an ANCOVA analysis might have been better suited than an ANOVA.

4. Why did TNF-a values increase in the TTE group to the authors opinion? If this is due to exercise intensity of the TTE, some more information on the particular exercises used should be given. However, it seems that there was no impact of TNF-a levels in either Pain or Disability levels.


5. Lines 81: Correct typo.

6. Line 121 & 133 & 138: Why are the 2 programs referred to as ‘back schools’? Please provide reasoning and reference(s). In no other spinal stabilization study, however, have they been referred as such.

7. Line 135: The study does not seem to be blind, as the authors imply. Please explain if otherwise. How were the purposes of the study conveyed to the participants, for instance.

8. Lines 159-161: The authors contend that initial treatment was not in the form of exercise, as the participants were in the acute stage of symptoms. However, from the inclusion criteria this cannot be surmised.

9. Line 180 & 181: The NRS used in the follow up is not quite the same as the VAS. It does not account for in-between decimal values, usually. For example, a patient might have indicated a pain level of 4 in the NRS, whereas in the VAS, a value of 3.6 might have been indicated. I understand this was the best available option over the telephone, but it is not correct to change the pain scale administration format as a trial progresses.

10. From lines 209-213 it is gathered that the study was not adequately powered, as a between-group difference in the VAS scale was not reached in either of the 3 follow up time points.

11. Lines 218-9: The term ‘Repeated ANOVA analyses’ does not specify which particular ANOVA type was used. A 2 x 4 perhaps for pain & disability?

12. Line 237: From figure 1 it can be seen that 5 & 4 participants left the TTE & SSE Groups respectively, therefore all participants did not complete the 20 sessions as mentioned in that particular text line. In fact, in line 228, this is clearly stated.

13. Line 294: One can have non-specific chronic LBP, also.

14. Figure 2 & 3: SD depiction seems not symmetrical around the mean value. Please amend or explain why.

15. Significant differences in Figure 3 are not depicted within the graph (by letter b, as indicated in Figure 2).

---

## Round 0.2 · Minor Revisions

I congratulate the authors for attending to all of the reviewers comments, with the very minor exception of the inclusion of some 95% confidence intervals ia highlighted by reviewer one. Please ensure these confidence intervals are included so that this manuscript can be accepted for publication in PeerJ.

Reviewer 1 ·

Basic reporting

Please see the "General comments for the author" section.

Experimental design

Please see the "General comments for the author" section.

Validity of the findings

Please see the "General comments for the author" section.

Additional comments

I want to thank the authors for their hard work to revise the manuscript. The quality of the manuscript is far more improved now.
I have just one minor comment. The authors mentioned that they have presented the standardized mean difference (SMD) with associate confidence intervals. However, I did not find the 95%CIs of the reported effect sizes in table 2. Please report them in Table 2.
Please rename the label in the 6th column as “Mean difference (95% CI)”. In addition, in column #7, please rename the label as “Effect size (95% CI)”

Annotated reviews are not available for download in order to protect the identity of reviewers who chose to remain anonymous.

Reviewer 2 ·

Basic reporting

Acceptable.

Experimental design

Reframed appropriately.

Validity of the findings

The scope of the article will ensure findings are considered appropriately moving forward.

Additional comments

The authors have reframed the study to an appropriate scope.

Reviewer 3 ·

Basic reporting

No comment

Experimental design

No comment

Validity of the findings

No comment

Additional comments

The authors have thoroughly revised thiw manuscript and it is now suitable for publication to Peer J.

---

## Round 0.3 · accepted · Accept

I thank the authors for their hard work attending to all of comments of the three reviewers and I. I am now happy to recommend this paper be accepted for publication in PeerJ.